

# Expression and prognosis of CDC45 in cervical cancer based on the GEO database

Zikang He[1], Xiaojin Wang[1], Zhiming Yang[2], Ying Jiang[1], Luhui Li[1], Xingyun Wang[1], Zheyao Song[1], Xiuli Wang[1,3], Jiahui Wan[4], Shijun Jiang[1,5], Naiwen Zhang[1] and Rongjun Cui[1]

[1] Department of Biochemistry and Molecular Biology, Mudanjiang Medical University, Mudanjiang, China
[2] Department of Clinical Laboratory, Handan Central Hospital, Handan, China
[3] Department of Clinical Laboratory, The Seventh Hospital in Qiqihar, Qiqihar, China
[4] Department of Clinical Laboratory, Harbin Public Security Hospital, Harbin, China
[5] Department of Clinical Laboratory, Daqing Medical College, Daqing, China

Corresponding author
Rongjun Cui,
cuirongjun@mdjmu.edu.cn

## ABSTRACT

Cervical cancer is one of the most common malignant tumors in women, and its morbidity and mortality are increasing year by year worldwide. Therefore, an urgent and challenging task is to identify potential biomarkers for cervical cancer. This study aims to identify the hub genes based on the GEO database and then validate their prognostic values in cervical cancer by multiple databases. By analysis, we obtained 83 co-expressed differential genes from the GEO database (GSE63514, GSE67522 and GSE39001). GO and KEGG enrichment analysis showed that these 83 co-expressed it mainly involved differential genes in DNA replication, cell division, cell cycle, etc.. The PPI network was constructed and top 10 genes with protein-protein interaction were selected. Then, we validated ten genes using some databases such as TCGA, GTEx and oncomine. Survival analysis demonstrated significant differences in CDC45, RFC4, TOP2A. Differential expression analysis showed that these genes were highly expressed in cervical cancer tissues. Furthermore, univariate and multivariate cox regression analysis indicated that CDC45 and clinical stage IV were independent prognostic factors for cervical cancer. In addition, the HPA database validated the protein expression level of CDC45 in cervical cancer. Further studies investigated the relationship between CDC45 and tumor-infiltrating immune cells *via* CIBERSORT. Finally, gene set enrichment analysis (GSEA) showed CDC45 related genes were mainly enriched in cell cycle, chromosome, catalytic activity acting on DNA, etc. These results suggested CDC45 may be a potential biomarker associated with the prognosis of cervical cancer.

## INTRODUCTION

Cervical cancer (CC) is one of the most prevalent gynecological malignancy in women, and its incidence and mortality are second only to breast cancer. It is estimated that 570,000 cases and 311,000 deaths from cervical cancer worldwide occurred in 2018 (*Bray et al., 2018*). The progression of CC takes approximately 10 to 20 years from a benign to a

malignant disease, with squamous cell carcinoma being its most common subtype (*Rajitha et al., 2021*). Despite significant advances in screening and various treatments, such as surgery, radiotherapy, and chemotherapy, deficiencies remain. Some studies indicated that over 90% of cases are caused by persistent infection with human papillomavirus (HPV), the main subtypes of which are HPV16 and HPV18 (*Schiffman et al., 2011*). The genetic sensitivity of CC is caused by HPV infection, which leads to genetic mutations. For instance, the polymorphism of GSTM1 is associated with high-risk HPV infection (*Lee et al., 2004*). In recent years, with the popularization and sharing of biomedical big data, the screening of effective molecular targets related to CC has become possible through bioinformatics methods. The previous studies have reported some targeted molecules regarding CC treatment (*Jiao et al., 2019*; *Wu et al., 2019*), but the clinical applications are very limited or even almost none. Therefore, an urgent and challenging task is to continue to explore early biomarkers in CC.

Cell division cycle (CDC45) is one of the proteins essential for the initiation and extension of DNA replication and for regulating DNA replication. It has been found that CDC45, mini-chromosome maintenance protein complex (MCM) and Go-Ichi-Ni-San (GINS) forms a "super complex" that is the central to eukaryotic replicons and has been shown to have helicase activity (*Masai, You & K-i, 2005*). It binds to DNA molecules and unwinds double-stranded DNA to form a replication fork structure throughout the entire DNA replication process (*Costa et al., 2011*; *Simon et al., 2016*). The previous studies reported that CDC45 may be a proliferation-associated antigen and contribute to the progression of malignant tumors (*Pollok et al., 2007*). However, the expression and function of CDC45 in CC still remains unknown.

In this study, we screened differentially expressed genes (DEGs) between cervical cancer tissues and normal or adjacent non-cancerous tissues based on the Gene Expression Omnibus (GEO) database, and collated and analyzed DEGs by a series of bioinformatics methods. Finally, we confirmed the important role of CDC45 in the development and prognosis of cervical cancer.

## MATERIALS AND METHODS

### Data collection and data processing

Using the keyword "cervical cancer" search on the GEO database (*Edgar, Domrachev & Lash, 2002*) (https://www.ncbi.nlm.nih.gov/geo/). The gene expression microarrays of GSE63514 (*Den Boon et al., 2015*), GSE67522 (*Sharma et al., 2015*; *Saha et al., 2017*), GSE39001 (*Espinosa et al., 2013*) and GSE52903 (*Medina-Martinez et al., 2014*) were downloaded. The GSE63514 dataset included 28 cancer tissues and 24 non-cancerous tissues. GSE67522 contained 20 cancer tissues and 22 non-cancerous tissues. GSE39001 contained 43 cancer tissues and 12 non-cancerous tissues. GSE52903 contained 55 cancer tissues and 17 non-cancerous tissues. Among them, the first three data sets are used as training sets and the last data set is used as a validation set. As the data come from the online database, no further approval from the Ethics Committee was required.

The differentially expressed genes (DEGs) between cancer tissues and non-cancerous tissues were screened out using GEO2R (*Davis & Meltzer, 2007*) (http://www.ncbi.nlm.nih.

gov/geo/geo2r). Probe sets with no corresponding gene symbols or genes with multiple sets of probe were removed or averaged, separately. |LogFC| > 1 and FDR < 0.05 were selected statistically significant (|logFC| stands for absolute value of the log fold change and FDR stands for false discovery rate). Co-expressed genes were obtained by intersection of DEGs from three datasets using Draw Venn Diagram (http://bioinformatics.psb.ugent.be/webtools/Venn/).

## GO and KEGG pathway enrichment analyses of DEGs

To identify DEGs associated pathways and function annotations, Gene Ontology (GO) and the Kyoto Encyclopedia of Genes and Genomes (KEGG) enrichment analyses were conducted by DAVID online database (*Huang, Sherman & Lempicki, 2009a*; *Huang, Sherman & Lempicki, 2009b*) (DAVID; https://david.ncifcrf.gov). GO is a widely used ontology in the field of bioinformatics, which covers three aspects of biology: biological process (BP), cellular component (CC), and molecular function (MF) (*Ashburner et al., 2000*). KEGG is one of the most commonly used bioinformatics tools in the world for understanding advanced functional and high-throughput experimental technologies of biological systems (*Kanehisa, 2002*). *P*-value < 0.05 indicated statistically significant difference.

## Construction of PPI network and screening of Key genes

The PPI network is to analyze the functional interactions between proteins using STRING online database, which is helpful to mine the core regulatory genes for the mechanisms of generation or development of diseases (*Franceschini et al., 2013*) (STRING; https://string-db.org/). In this study, we constructed to PPI network of DEGs and selected one interaction that was statistically significant with a composite score >0.4. Then, the PPI networks were mapped using Cytoscape 3.7.2 (*Smoot et al., 2011*) (https://cytoscape.org/), and the top 10 genes with the protein-protein interaction among the network were identified using CytoHubba which is a plug-in to Cytoscape.

## Validation of hub genes by the TCGA, GTEx and Oncomine databases

The transcriptome profiling counts data of CC were downloaded from The Cancer Genome Altas (TCGA) database (*Zhang et al., 2019*) (https://portal.gdc.cancer.gov/) with all subtypes of the project TCGA-CESC as inclusion criteria. Normal cervix tissues were downloaded from the Genotype-Tissue Expression (GTEx) database (*Battle et al., 2017*) (https://www.gtexportal.org/home/index.html). Data from both databases were combined and normalized. A total of 319 samples of CC included 306 cancer tissues and 13 adjacent non-cancerous tissues. We performed differential analysis of these samples and plotted the volcanic map and heat map using Perl (*Lindbom, Ribbing & Jonsson, 2004*) (http://www.perl.org/, version 5.32.0) and edgeR (*Robinson, McCarthy & Smyth, 2010*; *McCarthy, Chen & Smyth, 2012*) (http://bioinf.wehi.edu.au/edgeR/, version 4.0.2). |LogFC| > 1 and FDR < 0.05 were selected for statistical significance. Next, we performed overall survival (OS) analysis of 10 key genes using Gene Expression Profiling Interactive Analysis (GEPIA) tool (*Tang et al., 2017*) (http://gepia.cancer-pku.cn/), which is a new web server for analyzing the RNA sequencing expression data from the TCGA and the

GTEx databases. The expression level of hub genes in tumor and normal tissues was shown using GEPIA and Oncomine online database (*Rhodes et al., 2004*) (https://www.oncomine.org/resource/login.html).

## Cox proportional hazards regression analysis

Clinical information data on CC was downloaded from TCGA database. We analyzed the association with clinical information with genes expression by univariate and multivariate cox regression analysis and evaluated the influence of hub genes and clinicopathological factors on CC. $P$-value < 0.05 was set as the cut-off standard.

## ROC and DCA curve analysis

Receiver Operating Characteristics (ROC) curve analysis was performed based on pROC (*Robin et al., 2011*) and ggplot (https://ggplot2.tidyverse.org) packages in R software (version 4.0.2) for evaluating the sensitivity and specificity of CDC45 expression in CC diagnosis. The area under curve (AUC) is calculated to assess the veracity and reliability of diagnosis. Decision curve analysis (DCA) is a novel method for evaluating clinical outcome by comparing all-or-none clinical net-benefits (*Vickers & Elkin, 2006*; *Van Calster et al., 2018*). DCA curve was performed using ggDCA (https://CRAN.R-project.org/package=ggDCA) and survival (https://CRAN.R-project.org/package=survival) package in R software (version 4.0.2) based on clinical data from TCGA database.

## Analysis of CDC45 protein expression level by the HPA database

The Human Protein Atlas (HPA) database (*Uhlén et al., 2015*) (https://www.proteinatlas.org/) can spatially localize proteins at the single-cell level and detect more than 90% of the putative protein-coding genes. In the present study, we validate the protein level of CDC45 in normal cervix tissue and cervical cancer tissue by the HPA database.

## Relationship between CDC45 expression and tumor-infiltrating immune cells *via* CIBERSORT

CIBERSORT (*Gentles et al., 2015*) (http://cibersort.stanford.edu/) is a deconvolution algorithm based on gene expression, which estimates the $P$-value for deconvolution of each sample by Monte Carlo sampling, establishing a measure of confidence in the results. To explore the potential relationship between the CDC45 expression and tumor-infiltrating immune cells in CC, the mRNA expression matrix was standardized and the content of 22 human immune cells was calculated using CIBERSORT. We then divided the CC samples into two groups according to the median value and visualized the data using the vioplot (*Hu, 2020*) package in R software (version 4.0.2). $P$-value < 0.05 was considered statistically significant.

## Gene set enrichment analysis

Based on the correlation between gene pathways and CDC45 expression, we used Gene set enrichment analysis (*Subramanian et al., 2007*; *Subramanian et al., 2005*) (GSEA; https://www.gsea-msigdb.org/gsea/index.jsp) to generated a list of gene classifications and realized graphic visualization.

## RESULTS

### Screening out DEG in cervical cancer

Differentially expressed genes (DEGs) from the three datasets (GSE63514, GSE67522 and GSE39001) were identified using GEO2R standardizing gene microarray on the GEO database. Volcano plot and heatmap analysis showed that gene expression profiles from GSE63514 identified 4,608 differentially expressed genes with 3,053 up-regulated genes and 1,555 down-regulated genes in cervical cancer tissues when compared with normal cervical tissues. Gene expression profiles from GSE67522 identified 1327 differentially expressed genes with 588 up-regulated genes and 739 down-regulated genes. Gene expression profiles from GSE39001 identified 620 differentially expressed genes with 323 up-regulated genes and 297 down-regulated genes (Figs. 1A and 1B). Overlapping DEGs in the three datasets and plotting the Venn diagram (Fig. 1C) revealed that 83 co-expressed genes, among which 31 were highly expressed and 52 were low-expressed.

### GO and KEGG pathway enrichment analysis

GO and KEGG pathway enrichment analysis predicted the biological functions of DEGs. KEGG pathway analysis revealed that the DEGs were mainly enriched in DNA replication and cell cycle (Fig. 2A). GO enrichment analysis showed that changes in the biological processes (BP) of DEGs were significantly enriched in DNA replication, DNA-dependent DNA replication, and telomere maintenance *via* semi-conservative replication (Fig. 2B). Changes in cell component (CC) were mainly involved in condensed chromosome, chromosomal region, and centrosome (Fig. 2C). Genes associated with molecular function (MF) were mainly related to single-stranded DNA-dependent ATPase activity, DNA-dependent ATPase activity, and chemokine receptor binding (Fig. 2D).

### Construction of PPI network and screening of Hub genes

The PPI network of DEGs was constructed using STRING online database (Fig. 3A). The top 10 genes in the module with protein-protein interactions network were got using Cytoscape (Fig. 3B), including CDC45, RFC4, TOP2A, CCNA2, CCNB2, MCM6, KIF11, KIF20A, UBE2C and FEN1. We hypothesized they are hub genes that may play important roles in the development of CC.

### Verification and analysis of Hub genes in multiple databases

In order to validate above viewpoint, we collected mRNA sequencing data in CC from the TCGA and GTEx databases. Gene expression profiles identified 7,652 differentially expressed genes with 4,221 upregulated genes and 3431 downregulated genes (FDR < 0.05, |logFC| > 1). The heat map and volcano plot showed the distribution of differentially expressed genes (Figs. 4A and 4B). The survival analysis of 10 key genes showed that CDC45, RFC4 and TOP2A were of statistically significant (*P*-value < 0.05) (Fig. 5A and Fig. S1). Expression analysis by the Oncomine database and GEPIA showed that three genes were over-expressed in CC tissues compared with normal tissues (Figs. 5B and 5C), suggesting that the expression of CDC45, RFC4, and TOP2A was associated with prognosis of CC. According to clinical data from the TCGA database, univariate cox

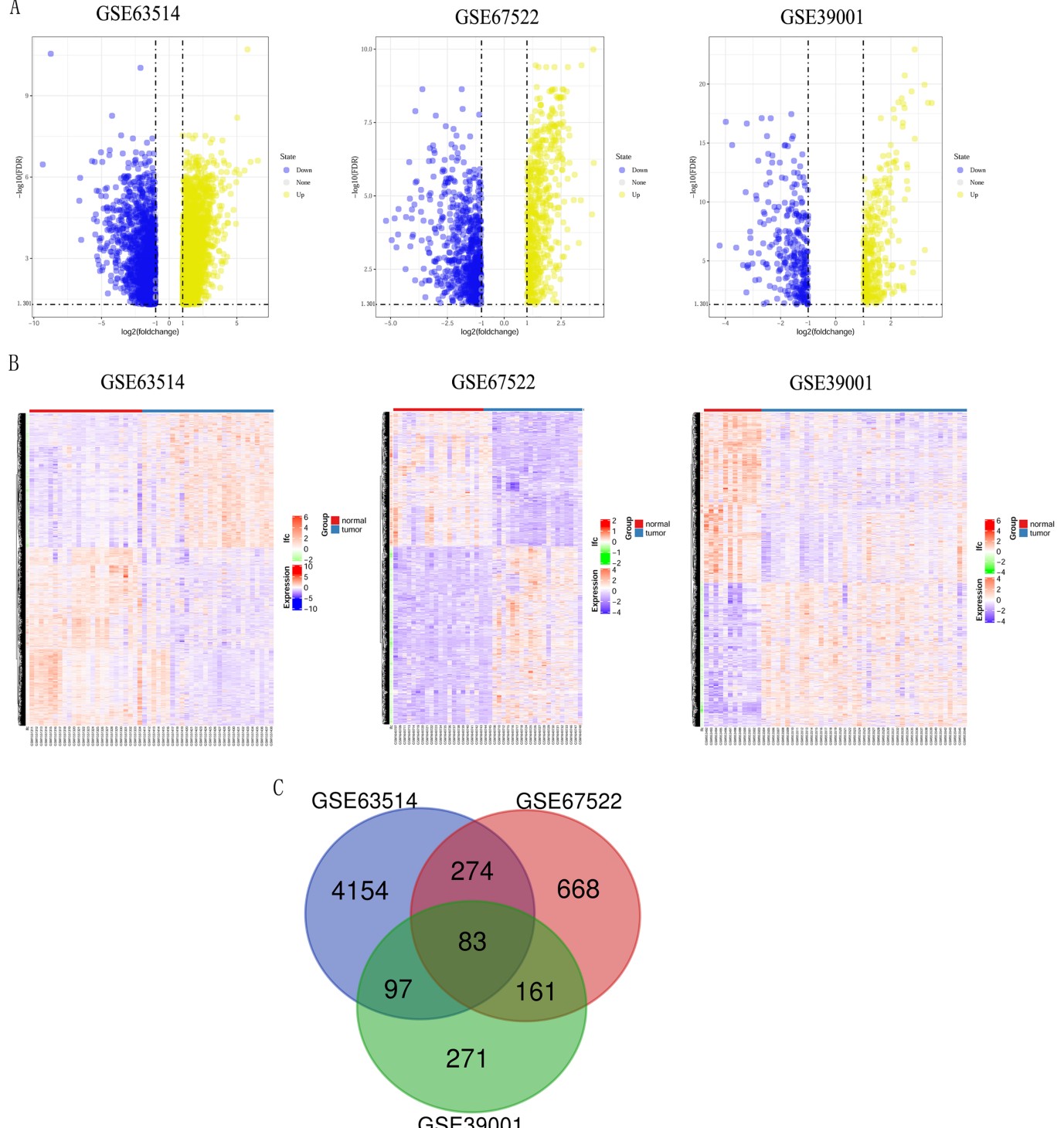

**Figure 1 Identification of differentially expressed genes in CC based on the GEO database.** (A) Volcano plot of the expression level of differentially expressed genes in normal and cancer tissues from GSE63514, GSE67522 and GSE39001. Yellow dots represent a high expression of genes and blue dots represent a low expression of genes. (B) Heatmap of the expression level of differential expressed genes between normal and cancer tissues from the three data sets. The abscissa indicates the sample names, and the ordinate shows the gene names. High expression of genes is shown in red and low expression of genes is shown in blue. LFC stands for log Fold Change. DEGs were defined with FDR < 0.05 (−log10 FDR > 1.301) and |logFC| > 1. (C) The three datasets showed an overlap of 83 genes using a Venn diagram.

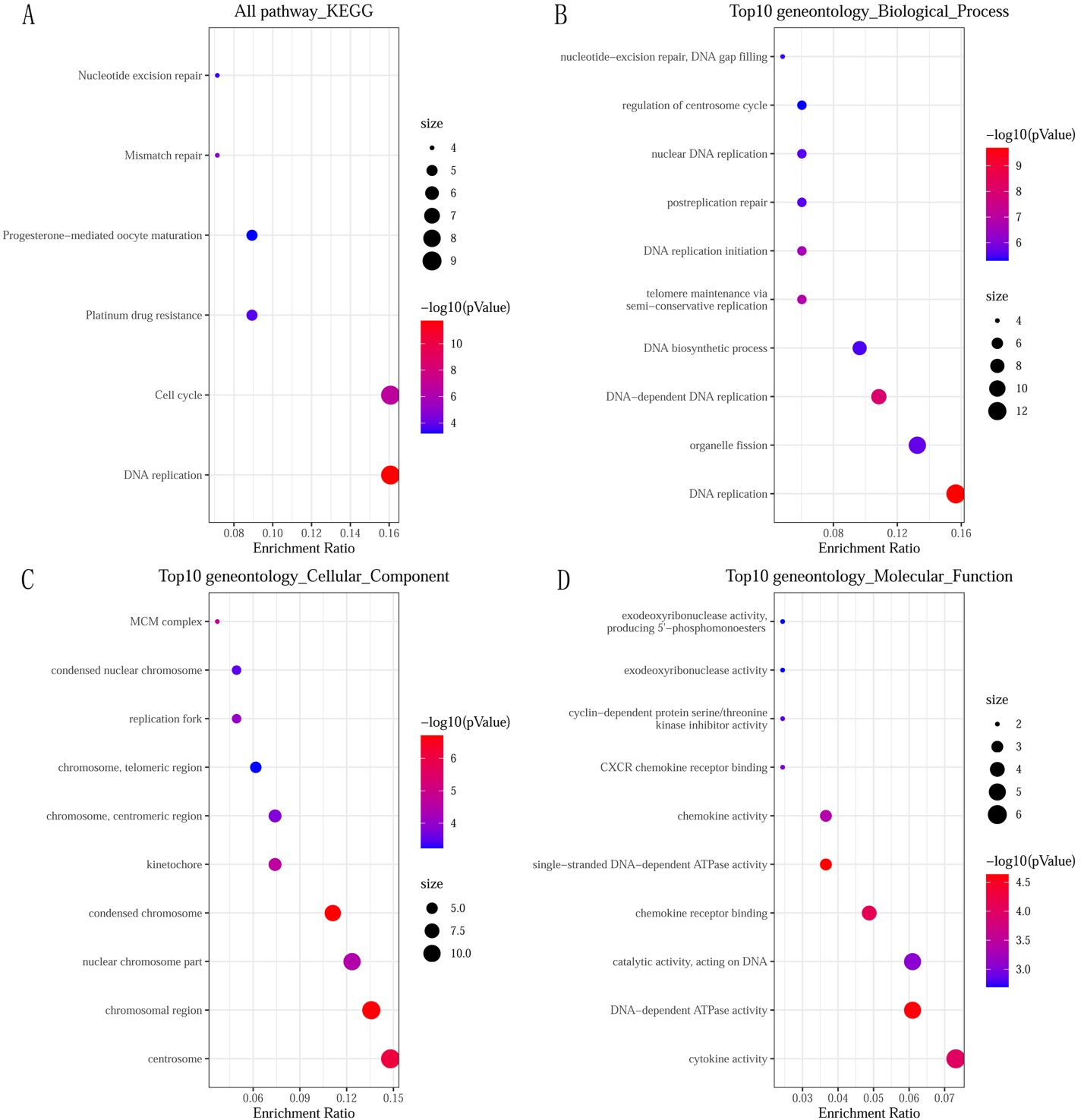

**Figure 2  Significantly functional enrichment pathway of 83 DEGs.** (A) KEGG pathway enrichment analysis. (B–D) the top 10 terms significantly enriched in the three GO categories: (B) biological process; (C) cellular component and (D) molecular function. *P*-value < 0.05 was set as the threshold.               

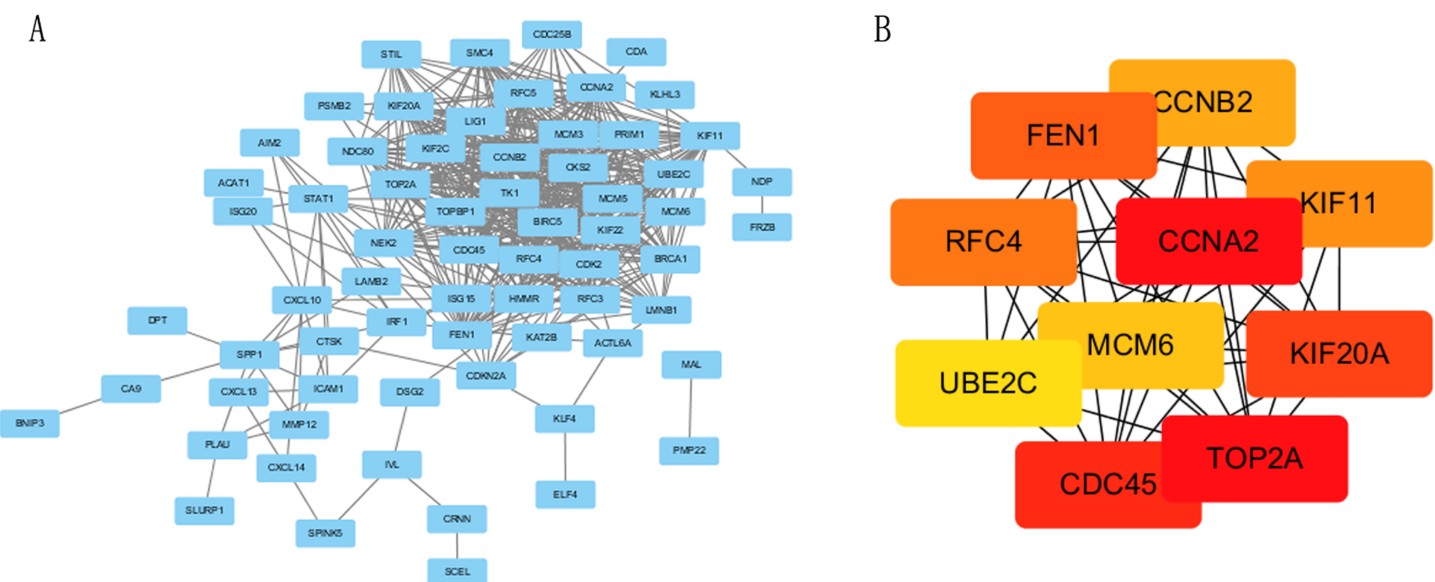

**Figure 3 PPI network and the most significant module of DEGs.** (A) The PPI network of DEGs was constructed using Cytoscape. (B) The most significant module was obtained from PPI network with 10 nodes and 200 edges.

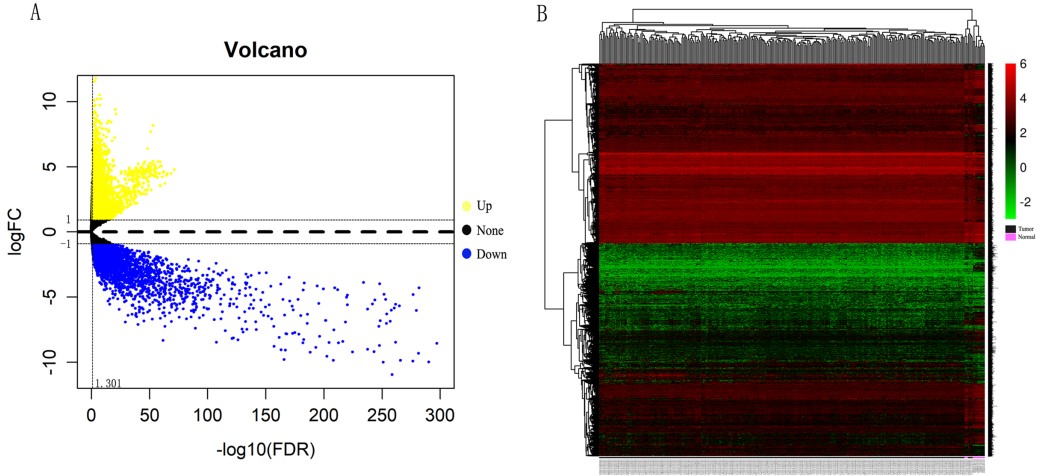

**Figure 4 Identification of differentially expressed genes in CC based on TCGA and GTEx databases.** (A) Volcano plot of the expression level of differentially expressed mRNAs in CC and adjacent normal tissues. Yellow dots represent a high expression of genes, black dots represent a normal expression of genes and blue dots represent a low expression of genes. (B) Heatmap of expression level of differentially expressed mRNAs between CC and adjacent normal tissues. The abscissa indicates the sample names, and the ordinate shows the gene names. Red represents high expression, and green represents low expression. DEGs were defined with FDR < 0.05 (−log10 FDR > 1.301) and |logFC| >1.

proportional regression analysis revealed that clinical stage IV and CDC45 are significantly associated with the development and progression of CC. Clinical stage IV and CDC45 were independent prognostic factors for CC in multivariate cox proportional regression analysis (Table 1 and Fig. 6).

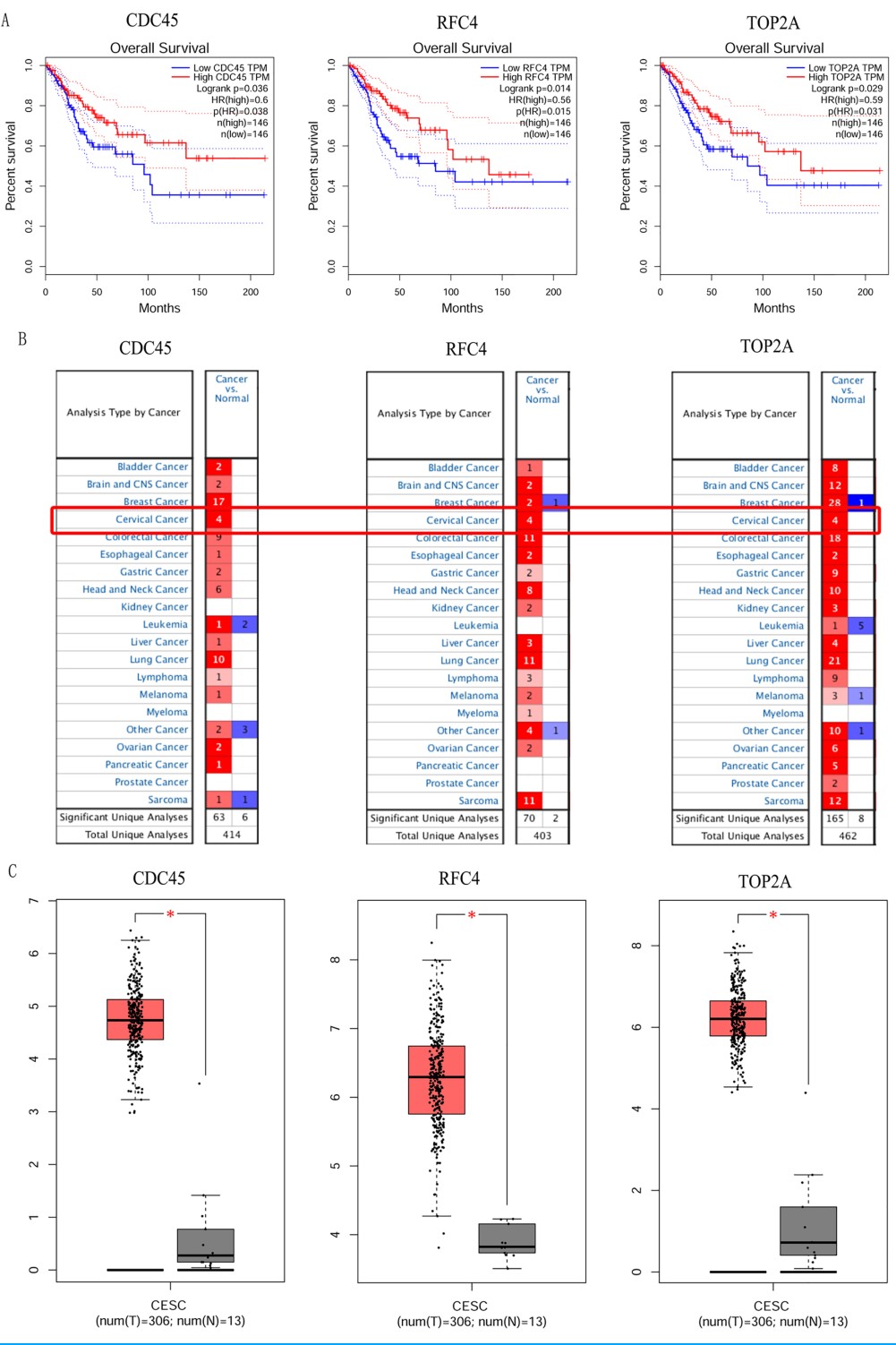

**Figure 5 Overall survival analysis and expression of hub genes in normal and cancer tissues.** (A) Survival curves analysis for CDC45, RFC4 and TOP2A. (B) The transcription levels of CDC45, RFC4 and TOP2A in the normal and cancer tissues (Oncomine). (C) Differential expression of CDC45, RFC4 and TOP2A in the normal and cancer tissues (GEPIA). * <0.05.

**Table 1 Association with overall survival and clinicopathologic characteristic in TCGA patients using Cox regression analysis.**

| Clinical characteristics | HR (95%CI) | *P*-val |
|---|---|---|
| **UniCOX** | | |
| Age | 1.02 [1.00–1.04] | 0.134 |
| Stage I | Ref. | |
| Stage II | 0.66 [0.28–1.51] | 0.318 |
| Stage III | 1.24 [0.58–2.66] | 0.582 |
| Stage IV | 6.22 [2.87–13.48] | <0.001*** |
| Grade 1 | Ref. | |
| Grade 2 | 1.02 [0.24–4.34] | 0.973 |
| Grade 3 | 1.00 [0.23–4.30] | 0.997 |
| CDC45 | 0.93 [0.88–0.98] | 0.008** |
| RFC4 | 0.99 [0.96–1.01] | 0.203 |
| TOP2A | 1.00 [0.99–1.02] | 0.598 |
| **MultiCOX** | | |
| Stage I | Ref. | |
| Stage II | 0.59 [0.25–1.37] | 0.220 |
| Stage III | 1.12 [0.52–2.42] | 0.766 |
| Stage IV | 5.54 [2.55–12.04] | <0.001*** |
| CDC45 | 0.93 [0.88–0.98] | 0.012* |

Note:
* <0.05.
** <0.01.
*** <0.001.
HR, Hazard Ratio; Ref, reference group.

## ROC and DCA curve analysis on CDC45

Based on CDC45 expression in CC from three datasets of the GEO database (GSE63514, GSE67522 and GSE39001), we performed ROC curve analysis separately to assess its specificity for CC diagnosis. The cut-off values were 10.281 for GSE63514, 335.486 for GSE67522 and 5.362 for GSE39001. The AUC of all three data sets was > 0.8 (Figs. 7A–7C), indicating that CDC45 has significant sensitivity and specificity for CC diagnosis. Similarly, DCA curve has also shown that the net benefit of CDC45 exceeds that of the reference model over the entire range of thresholds (Fig. 7D). These results suggested that CDC45 could be used as a potential biomarker for the diagnosis of CC.

## Validation of CDC45 by the GEO database

To further validate CDC45 expression in CC, we screened a gene microarray data set of CC (GSE52903) from the GEO database for differential expression analysis and plotted volcano and heatmap. The results found that CDC45 expression remained significant (Figs. S2A and S2B). Box plots were plotted using the unpaired t-test method by GraphPrism software. ROC curve analysis showed that the cut-off value was 0.721 and the

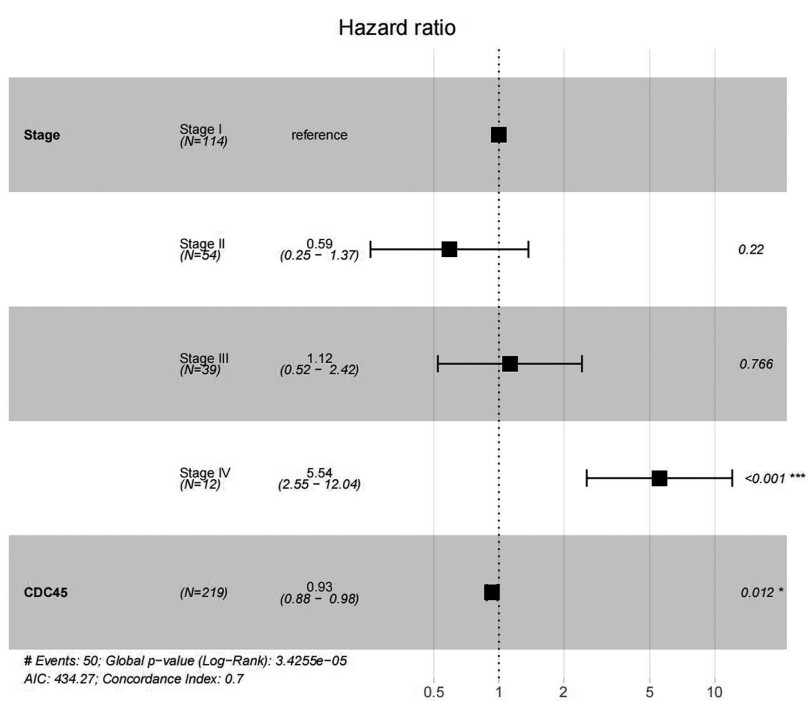

**Figure 6 Multivariate Cox analysis of CDC45 expression and clinical stage.**

AUC was 0.9. These results suggested that CDC45 was significant in the development of CC (Figs. S2C and S2D).

## Protein expression level of CDC45 on the HPA database

Immunohistochemistry (IHC) staining obtained by the HPA database demonstrated the expression status of the CDC45 and the patient clinical data (Fig. 8). The result showed that the protein expression level of CDC45 was positively correlated with disease status and it was up-regulated in CC tissue, which suggested that the effect of CDC45 that we found as reliable.

## Correlation analysis between CDC45 expression and tumor-infiltrating immune cells in CC

Previous studies suggest that tumor-infiltrating lymphocytes are independent predictors of sentinel lymph node status and survival in cancers (Azimi et al., 2012). Therefore, we investigated whether CDC45 expression was associated with tumor-infiltrating immune cells in CC using CIBERSORT. Figs. 9A and 9B showed that the relative content distribution of 22 immune cells and the correlation between 22 immune cells in CC. Then, we divided 306 CC samples into high and low groups based on the median value, and calculated the difference and correlation of CDC45 expression in 22 immune cells. The results showed that activated memory CD4+ T cells ($R = 0.28$, $P = 3.2e{-}05$) and follicular helper T cells ($R = 0.2$, $P = 0.0026$) were positive relation with the expression of

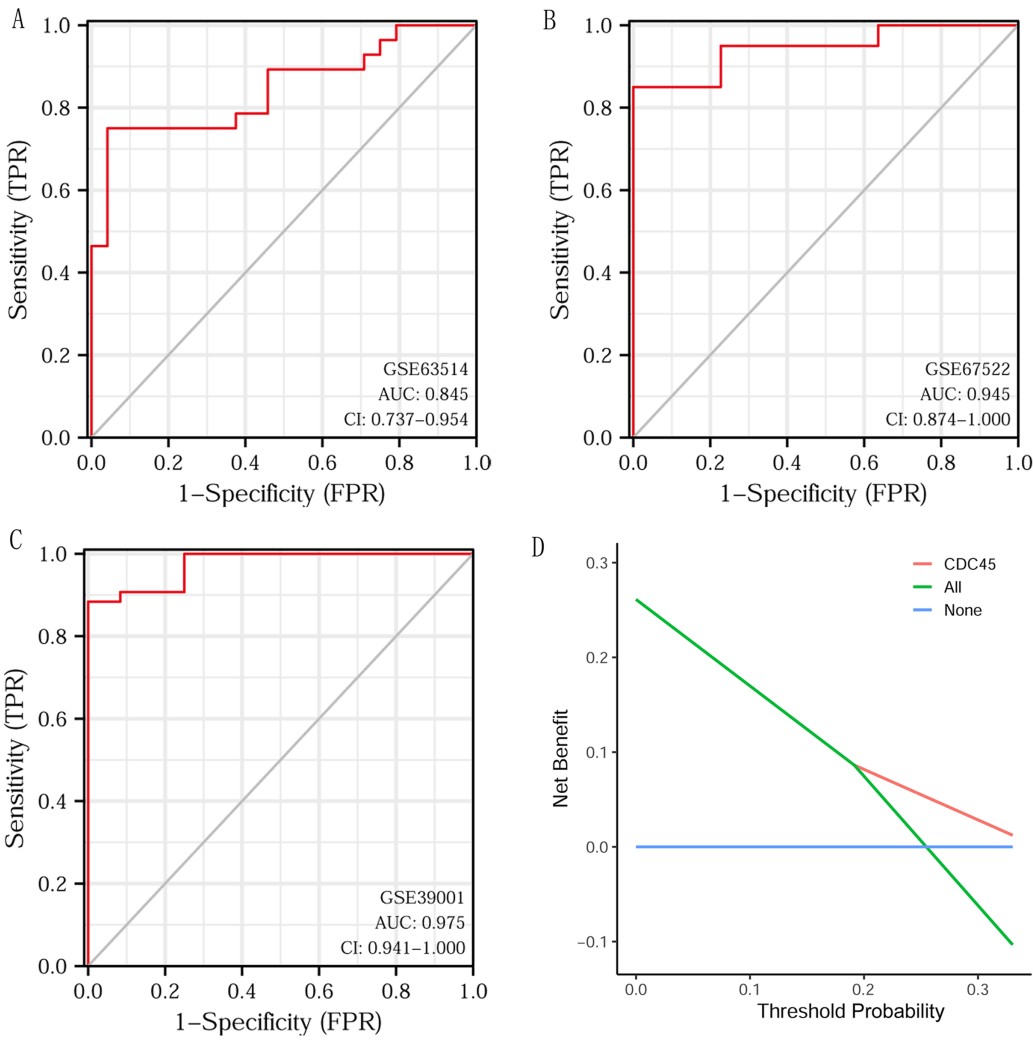

**Figure 7 ROC and DCA curve analysis.** (A–C) ROC curve analysis of CDC45 on the GEO database (GSE63514, GSE67522 and GSE39001). (D) The decision curve analysis (DCA) shows the net benefit of CDC45 in the 3-year survival. The abscissa represented the threshold probabilities, and the ordinate measured the net benefit.

CDC45. Naive B cell ($R = −0.19$, $P = 0.0044$) and resting memory CD4+ T cells ($R = −0.19$, $P = 0.0049$) were negative relation with the expression of CDC45 (Figs. 9C and 9D).

## Gene sets enriched in CDC45 expression phenotype

CDC45 related signaling pathways were analyzed base on GSEA to identify the signaling pathways with significant differences (FDR < 0.05, NOM $P$-value < 0.05) in GO and KEGG enrichment of the highly expression data sets in CC (Table 2).

Five KEGG items including purine metabolism, cell cycle, oocyte meiosis, pyrimidine metabolism, DNA replication were showed significantly differential enrichment in the CDC45 high expression phenotype (Fig. 10). GO items results displayed that the biological process of the CDC45 high expression phenotype was mainly enriched in the chromosome, nuclear chromosome, chromosome region, catalytic complex, and

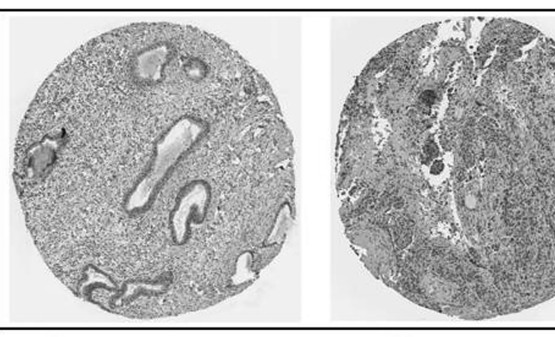

Cervix, uterine
Patient ID: 1657
Sex: female
Age: 70
Stain: Not detected
Intensity: weak
Quantity: <25%

CC
Patient ID: 926
Sex: female
Age: 54
Stain: Low
Intensity: weak
Quantity: >75%

**Figure 8 Analysis of the protein expression level of CDC45 in CC by the Human Protein Atlas (HPA) database.**

microtubule cytoskeleton. The cellular component of the CDC45 high expression phenotype was mostly enriched in the cell cycle, cell cycle process, DNA replication, DNA metabolic process, and cellular response to DNA damage stimulus. The molecular function of the CDC45 high expression phenotype was chiefly enriched in catalytic activity action on DNA, chromatin binding, ubiquitin-like protein binding, ATPase activity, and hydrolase activity action on acid anhydrides.

## DISCUSSION

Cervical cancer (CC) is the fourth most common malignant tumor in women. It has been reported that there was a high mortality rate owing to cervical cancer worldwide (*Bray et al., 2018*). In recent years, the incidence of cervical cancer is younger and younger, resulting in a shorter life expectancy (*Li, Wu & Cheng, 2016*; *Kong et al., 2019*). Although the clinical therapy of cervical cancer has achieved substantial progress, it remains high in the advanced mortality rate. Therefore, it is an extremely urgent that identify the potential biomarkers for cervical cancer and clinical treatment and prognosis.

In our present study, we investigated whether CDC45 has a potentially risk effect on CC. To elucidate the effect of CDC45, we performed a large number of data mining and analysis by some online databases to detect expression levels of CDC45 in CC. We found CDC45 was highly expressed in CC. These results showed that high expression of CDC45 may play a crucial role in the progression and prognosis of CC.

Here, we first found that 83 candidate genes of cervical cancer are mainly enriched in DNA replication, cell cycle and cell division by GO and KEGG pathway analysis. This suggested that the development of cervical cancer may be related to abnormal changes in the cell cycle. Survival analysis was performed on the top 10 genes with protein-protein interactions showed that the overall survival (OS) time of CDC45, RFC4 and TOP2A

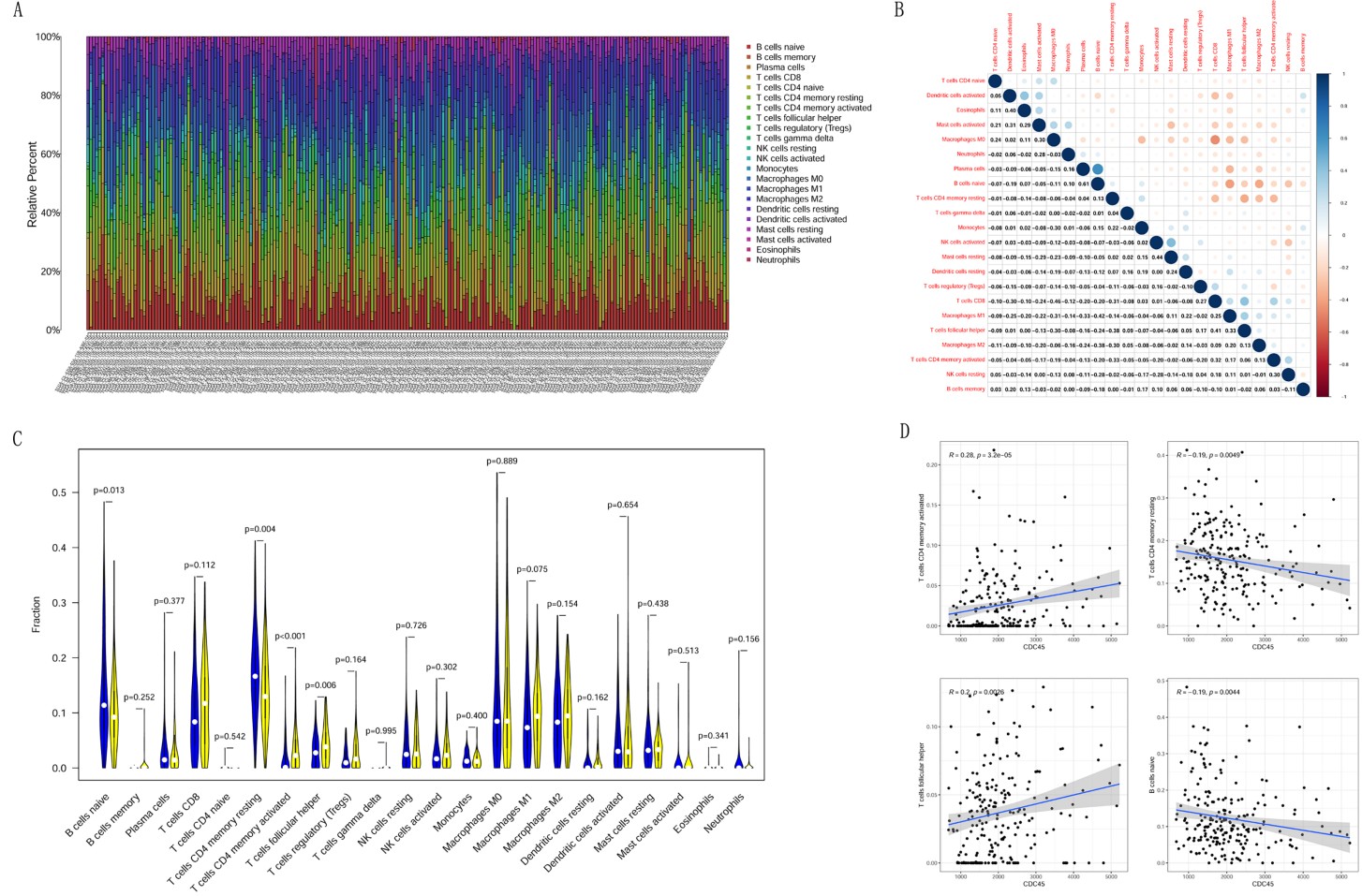

**Figure 9 The relationship between the CDC45 expression and tumor-infiltration immune cells.** (A) Barplot showed the relative content of 22 immune cells in CC samples. (B) Block diagram showed the correlation of 22 immune cells in CC. (C) Violin diagram showed the difference of CDC45 expression in 22 immune cells. High expression groups are indicated in yellow and low expression groups in blue. (D) Scatterplot showed the correlation between CDC45 and immune cells.

was significantly correlated with the prognosis of cervical cancer. Indeed, we found that the expression of three genes was up-regulated in cervical cancer tissues compared with normal or adjacent tissues through the validation with multiple databases. Furthermore, we performed univariate and multivariate cox regression analysis in combination with the clinical information of the patients. The results showed that the CDC45 and clinical stage IV of cervical cancer were significantly associated with cervical cancer, suggesting that CDC45 and clinical stage IV may be independent risk factors for the development of cervical cancer. The ROC and DCA curve analysis were also consistent with the above results. Moreover, CDC45 is common highly expressed in pan-cancer (Fig. S3).

In recent years, cancer immunotherapy is a hot topic in cancer treatment (*Bader, Voss & Rathmell, 2020*; *Yang, 2015*). We therefore investigated whether the CDC45 expression was associated with tumor-infiltration immune cells in CC. Interestingly, expression of CDC45 was positively associated with activated memory CD4+ T cells and T follicular helper cells. Previous studies have suggested that dysregulation of memory CD4 + T cells is

**Table 2 Gene sets enriched in phenotype.**

| Gene set name | NES | NOM p-val | FDR Q-val |
|---|---|---|---|
| **KEGG** | | | |
| KEGG_PURINE_METABOLISM | −2.178 | 0 | 6.14E−04 |
| KEGG_CELL_CYCLE | −2.158 | 0 | 3.07E−04 |
| KEGG_OOCYTE_MEIOSIS | −2.101 | 0 | 5.53E−04 |
| KEGG_PYRIMIDINE_METABOLISM | −2.075 | 0.002 | 0.001 |
| KEGG_DNA_REPLICATION | −2.067 | 0 | 8.28E−04 |
| **GO_BP** | | | |
| GO_CHROMOSOME | −2.321 | 0 | 5.57E−04 |
| GO_NUCLEAR_CHROMOSOME | −2.301 | 0 | 2.79E−04 |
| GO_CHROMOSOMAL_REGION | −2.184 | 0 | 5.57E−04 |
| GO_CATALYTIC_COMPLEX | −2.145 | 0.002 | 8.18E−04 |
| GO_MICROTUBULE_CYTOSKELETON | −2.140 | 0 | 6.54E−04 |
| **GO_CC** | | | |
| GO_CELL_CYCLE | −2.460 | 0 | 0 |
| GO_CELL_CYCLE_PROCESS | −2.409 | 0 | 0 |
| GO_DNA_REPLICATION | −2.375 | 0 | 0 |
| GO_DNA_METABOLIC_PROCESS | −2.372 | 0 | 0 |
| GO_CELLULAR_RESPONSE_TO_DNA_DAMAGE_STIMULUS | −2.369 | 0 | 0 |
| **GO_MF** | | | |
| GO_CATALYTIC_ACTIVITY_ACTING_ON_DNA | −2.097 | 0 | 0.010 |
| GO_CHROMATIN_BINDING | −2.093 | 0 | 0.005 |
| GO_UBIQUITIN_LIKE_PROTEIN_BINDING | −2.092 | 0 | 0.003 |
| GO_ATPASE_ACTIVITY | −2.091 | 0 | 0.002 |
| GO_HYDROLASE_ACTIVITY_ACTING_ON_ACID_ANHYDRIDES | −2.085 | 0 | 0.002 |

promoting the progression of malignancy (*Gasper, Tejera & Suresh, 2014*; *MacLeod et al., 2009*). The crucial role of T follicular helper cells is to help B cells produce antibodies and participate in humoral immunity (*Crotty, 2019*). Our researches indicated that activated memory CD4 + T cells and T follicular helper cells have better prognostic value in patients with CC consistent with previous results. However, here is a limitation that further studies are needed to illustrate the molecular characters of activated memory CD4 + T cells and T follicular helper cells to explain their prognostic potential.

CDC45 can act as a DNA replication initiation factor (*Hennessy et al., 1991*). It was first proposed in 1997 that the gene has a genetic correlation with the DNA replicators MCM5/CDC46, MCM7/CDC47 and ORC genes previously discovered, and is specifically related to the stability of G1/S mRNA (*Zou, Mitchell & Stillman, 1997*; *Hardy, 1997*; *Hopwood & Dalton, 1996*). CDC45 is believed to be involved in the development and progression of different tumors and serves as a potential therapeutic target. For instance, *Huang et al. (2019)* found that the low expression of CDC45 can suppress cell proliferation in non-small cell lung cancer (NSCLC), resulting in the cell were stagnated in G2/M phase of cell cycle. This result shows that CDC45 is supporting the carcinogenic effects.

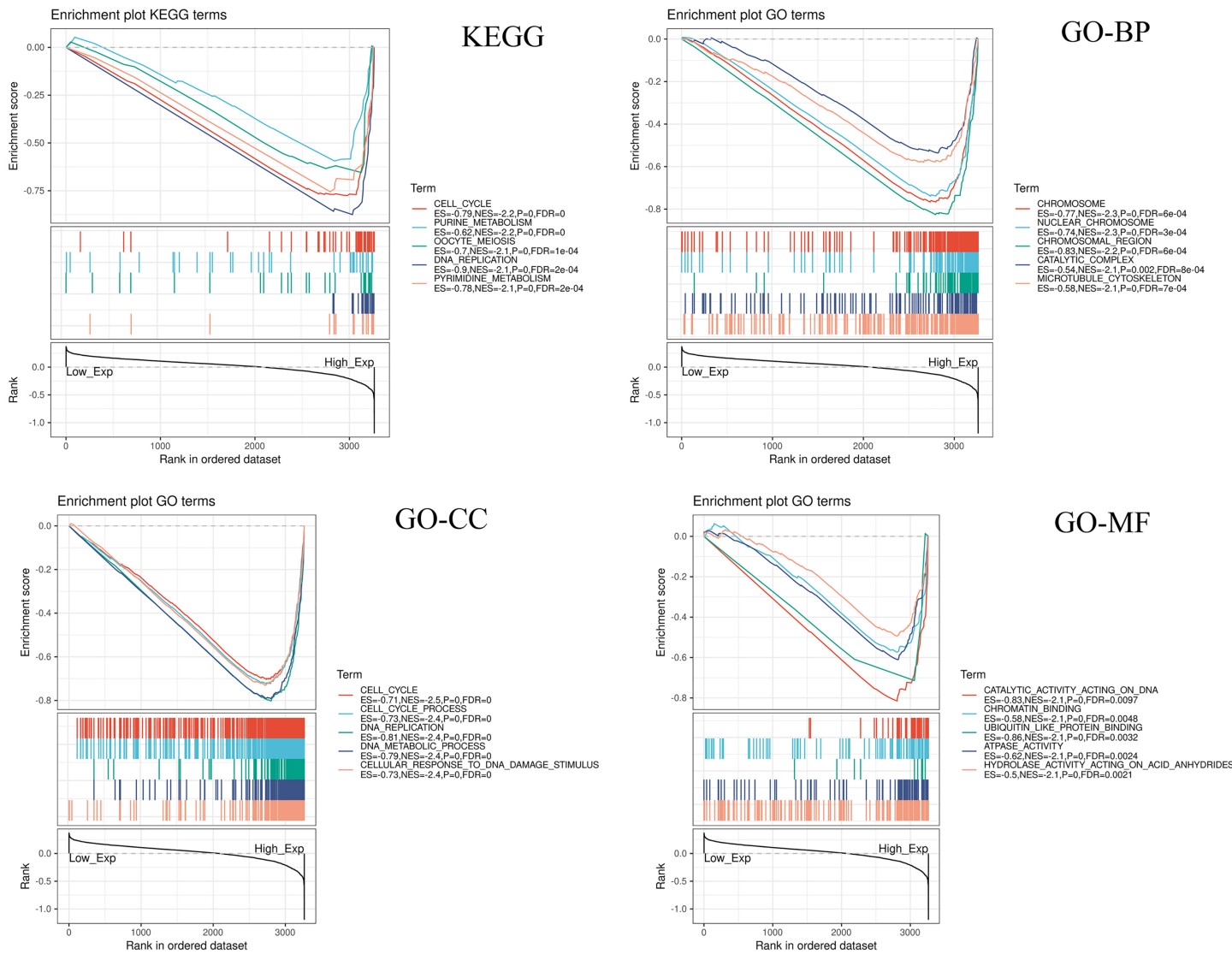

**Figure 10 Enrichment plots from gene set enrichment analysis (GSEA).** Differential enrichment of gene in KEGG, GO-BP, GO-CC and GO-MF pathways with high CDC45 expression. (KEGG, Kyoto Encyclopedia of Genes and Genomes; GO, Gene Ontology; BP, biological process; CC, cellular component; MF, molecular function).

Sun et al. found that the expression of CDC45 was up-regulation in papillary thyroid cancer (PTC), which promoted the proliferation of cancer cell *in vitro* and tumor growth *in vivo* (*Sun et al., 2017*). To confirm the function of the CDC45 in cervical cancer, we performed single gene enrichment analysis by GSEA. The results showed that purine metabolism, cell cycle, oocyte meiosis, pyrimidine metabolism, and DNA replication in KEGG, chromosome, nuclear chromosome, chromosome region, catalytic complex, and microtubule cytoskeleton in the biological process of GO, cell cycle, cell cycle process, DNA replication, DNA metabolic process and cellular response to DNA damage stimulus in the cell cycle of GO, catalytic activity acting on DNA, chromatin binding, ubiquitin-like protein binding, ATPase activity and hydrolase activity acting on acid anhydrides in the molecular function of GO are significantly enriched in CDC45 high expression

phenotype. However, those pathways are no significantly enriched in the CDC45 low expression phenotype (no show). The results indicated that highly expressed CDC45 can be used as a potential biomarker of prognosis and therapeutic target in CC patients. Furthermore, previous studies have also found that CDC45 was indeed associated with prognosis in cervical cancer, which strongly demonstrates the reliability of our results (*Qiu et al., 2020*).

In this paper, we acknowledged our researches are only limited to mining and analysis of the online database without wet experiment verification. For instance, CDC45 expression in CC was examined at the cellular level by the methods of real-time PCR, and MTT, and so on. We strongly recommend further research in this area to increasing evidence for the biological effect of CDC45.

# CONCLUSIONS

Based on GEO and other multi-database biological big data mining, we found that CDC45 can be involved in the development of cervical cancer as an independent prognostic factor. This study provides a new potential target for the clinical diagnosis of cervical cancer. Meanwhile, the relationship between the CDC45 expression and immune-infiltrating cells suggests that immunotherapy may facilitate the treatment of cervical cancer.

# ACKNOWLEDGEMENTS

We acknowledge TCGA and GEO database for providing their platforms and contributors for uploading their meaningful datasets.

## Funding

This work was supported by grants from the Postgraduate Innovation Research Project of Mudanjiang Medical University (NO.YJSCX-MY04). The funders had no role in study design, data collection and analysis, decision to publish, or preparation of the manuscript.

## Grant Disclosures

The following grant information was disclosed by the authors:
Postgraduate Innovation Research Project of Mudanjiang Medical University: YJSCX-MY04.

## Competing Interests

The authors declare that they have no competing interests.

## Author Contributions

- Zikang He conceived and designed the experiments, performed the experiments, prepared figures and/or tables, authored or reviewed drafts of the paper, and approved the final draft.

- Xiaojin Wang conceived and designed the experiments, performed the experiments, prepared figures and/or tables, authored or reviewed drafts of the paper, and approved the final draft.
- Zhiming Yang conceived and designed the experiments, performed the experiments, prepared figures and/or tables, authored or reviewed drafts of the paper, and approved the final draft.
- Ying Jiang performed the experiments, analyzed the data, authored or reviewed drafts of the paper, and approved the final draft.
- Luhui Li performed the experiments, analyzed the data, authored or reviewed drafts of the paper, and approved the final draft.
- Xingyun Wang performed the experiments, analyzed the data, authored or reviewed drafts of the paper, and approved the final draft.
- Zheyao Song performed the experiments, analyzed the data, authored or reviewed drafts of the paper, and approved the final draft.
- Xiuli Wang performed the experiments, analyzed the data, authored or reviewed drafts of the paper, and approved the final draft.
- Jiahui Wan performed the experiments, analyzed the data, prepared figures and/or tables, authored or reviewed drafts of the paper, and approved the final draft.
- Shijun Jiang performed the experiments, prepared figures and/or tables, authored or reviewed drafts of the paper, and approved the final draft.
- Naiwen Zhang performed the experiments, authored or reviewed drafts of the paper, and approved the final draft.
- Rongjun Cui conceived and designed the experiments, performed the experiments, authored or reviewed drafts of the paper, and approved the final draft.

### Data Availability
The data is available at NCBI GEO: GSE63514, GSE67522, GSE39001, GSE52903.

The R code is available upon request.

### Supplemental Information
Supplemental information for this article can be found online at http://dx.doi.org/10.7717/peerj.12114#supplemental-information.

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
