# Peer review of "Expression and prognosis of CDC45 in cervical cancer based on the GEO database"

_PeerJ, doi:10.7717/peerj.12114_

## Round 0.1 · original submission · Major Revisions

Thank you for submitting your work to Peer J. Your manuscript has been peer-reviewed. It is interesting. However, you must carefully revise your manuscript per the reviewers' comments.

Reviewer 1 ·

Basic reporting

1. Please properly cite any software and databases used in the analysis (including GEO, GEO2R, GO, KEGG, R, ggplot2 and etc) if possible.
2. In Line 87, please be specific on how you downloaded the 257 samples on the TCGA website including inclusion and exclusion criteria. Did you use "CESC" as the keyword? and etc.
3. The non-cancerous tissue samples are too few to compare to (n = 2). Is it possible that any additional samples can be added?
4.

Here are the comments regarding the figures and tables.
1. Please add the value corresponding to the vertical and horizontal cutoff in Fig 1A. The legend in Fig 1B is illegible.
2. Please add the cutoff lines in Fig 5A. The legend in Fig 5B lacks annotation for expression levels and the labels for both axes. Please add the sample size for each group in Fig 5C.
3. Where are there only two variables in the multivariate cox regression shown in Table 1? Please include the comparison and the control group for each categorical clinical variable including stage and grade. Why doesn't the age variable get included in the univariate analysis?


Below lists a list of minor suggestions.
1. In Line 29, there is an additional space between "We" and "utilized".
2. In Line 31 and 95, what is "cox logistic regression analysis"? Do you mean "cox regression"?
3. In Line 93, capitalized "c" in "clinical".
4. Abbreviation is missing for TCGA, FC, FDR.
5. Please include the R version in Line 104.

Experimental design

1. We highly recommend sharing any codes (for example, R scripts) used in the analysis for scientific reproducibility.

Validity of the findings

1. Given that the GEPIA consists of TCGA data, the analysis from line 144 wouldn't be considered as an independent verification of Hub genes. Please read more here http://gepia.cancer-pku.cn/about.html.

Reviewer 2 ·

Basic reporting

There are a lot of grammatical errors and typos in this manuscript. For examples:
- This study aims to identified the hub genes ...
- ... the module of top 10 genes with the most protein interaction were selected.
- The expression level of hub genes were shown ...
- No space before dot or comma.
- The use of inconsistent capital and small letters.
- ...
Thus the authors are suggested to re-check and revise carefully.

The manuscript lacks detailed information on problem background, and especially literature reviews. No study on cervical cancer-based bioinformatics analysis or CDC45 gene is discussed.

Experimental design

Did the authors concern about batch effect removal when merging three databases?

Source codes should be provided for replicating the methods.

ROC curves and AUCs have been used in previous bioinformatics studies i.e., PMID: 31750297, PMID: 33735760, and PMID: 32613242. Therefore, the authors are suggested to refer to more works in this description to attract a broader readership.

Validity of the findings

Why multivariate analysis did not generate a signature gene according to risk scores?

The authors should have some validation data (maybe from TCGA or other sources).

What are the limitations of this study? It should be discussed.

Additional comments

No comment.

---

## Round 0.2 · accepted · Accept

I am writing to inform you that your manuscript - Expression and prognosis of CDC45 in cervical cancer based on the GEO database - has been Accepted for publication. Congratulations!

Reviewer 1 ·

Basic reporting

No comment

Experimental design

No comment

Validity of the findings

No comment

Additional comments

Thank you so much for taking the time to address all my comments and re-analyzing the data. Please make sure that you include a note saying that "The code is available upon request from the authors" given that you won't attach the R code in the supplemental.

Reviewer 2 ·

Basic reporting

No comment.

Experimental design

No comment.

Validity of the findings

No comment.

Additional comments

No comment.